# Improving Energy Harvesting from Bridge Vibration Excited by Moving Vehicles with a Bi-Stable Harvester

**DOI:** 10.3390/ma15062237

**Published:** 2022-03-17

**Authors:** Zhiyong Zhou, Haiwei Zhang, Weiyang Qin, Pei Zhu, Wenfeng Du

**Affiliations:** 1School of Civil Engineering and Architecture, Henan University, Kaifeng 475004, China; 10160091@vip.henu.edu.cn (Z.Z.); hwzhang@henu.edu.cn (H.Z.); 10160016@vip.henu.edu.cn (W.D.); 2School of Power and Energy, Northwestern Polytechnical University, Xi’an 710072, China; 3Department of Engineering Mechanics, Northwestern Polytechnical University, Xi’an 710072, China; qinweiyang@nwpu.edu.cn

**Keywords:** energy harvesting, bridge vibration, bi-stable energy harvester, linear energy harvester, snap-through motion

## Abstract

To monitor the health status of the bridge, many sensors are needed to be mounted on it. Converting bridge vibration energy to electrical energy is considered as a potential solution to the problem of providing reliable electric power to these sensors. The objective of this work is to present an operable strategy for improving the electric energy output of a piezoelectric energy harvester installed on a bridge by introducing bi-stable characteristics. A bi-stable harvester is proposed. By adjusting the tip and fixed magnets, different types of potential energy can be generated, and then the harvester can exhibit the linear, mono-stable and bi-stable characteristics. In the bi-stable state, the harvester triggers snap-through motions easily and generates large outputs. The corresponding prototype was fabricated, and the experiment was carried out to validate the advantage of the bi-stable energy harvester. The experiment results show that the bi-stable energy harvester outperforms the classical linear harvester over the whole range of vehicle speed. As the vehicle speed exceeds a critical one, the snap-through motion will happen, which is beneficial to enhancing the electricity output. This conceptual design may provide guidance for promoting the performance of bridge energy harvesting.

## 1. Introduction

Bridges are becoming a more and more important part of the modern transportation system [1]. It is necessary to monitor the health of bridges at all times to ensure their safety. Although deploying wireless sensors in the detection region is an ideal scenario to monitor the state of the bridge, providing cheap and reliable power to the sensors is a well-known difficulty [2]. At present, the conventional electrochemical battery is still the primary power source of wireless sensors, but replacing the battery costs a lot, especially in dangerous locations [3]. Moreover, the heavy metals in wasted batteries pose a threat to the health and living environment of human beings. Energy harvesting from vibrations is considered a great potential energy solution to powering wireless sensors. In practice, there exists abundant vibration energy when vehicles pass through the bridge. This vibration energy results from passing vehicles and is not affected by the weather, geographical conditions and other factors, so it emerges as a promising solution for the power of bridge health monitoring systems [4]. Moreover, the piezoelectric energy harvester is usually simple in structure and easy to be manufactured. Thus, it can be produced in a large amount, so the vibration-based energy piezoelectric harvester has received more and more attention [5].

In recent decades, there are many studies focusing on converting the energy of bridge vibrations into electric energy. Ali et al. [6] investigated a linear energy harvester for the highway bridge under the excitation of a moving vehicle. The highest efficiency of energy harvesting was sensitively affected by the resonance frequency. Peigney and Siegert [7] investigated energy harvesting from a concrete highway bridge by a cantilever energy harvester. The results showed that the electric energy output could reach 0.03 mW. Zhang et al. [8] used a cantilever beam energy harvester to harvest energy from four bridges excited by truck models. The results showed that poorer road conditions on bridges were helpful for harvesting vibration energy. Karimi et al. [9] considered inertial effects on energy harvesting from a bridge with a heavy vehicle traveling on it. Zhang et al. [10] analyzed the output voltages of the piezoelectric energy harvester for nine different conditions. The vehicle speed had a significant effect on the energy output. Since the bridge response is the excitation source of the linear piezoelectric energy harvester (LPEH), the resonant frequency of LPEH must be tuned to the fundamental frequency of the bridge to obtain a high harvesting efficiency. Thus, some significant efforts have been devoted to studying the vibration response of a bridge and estimating the fundamental frequency and stiffness parameters. Aloisio et al. [11] conducted a modal analysis for a multi-span concrete bridge to estimate the compressive strength of concrete specimens with seven spans. Bonopera et al. [12] predicted the short-term dynamics of the uncracked prestressed concrete members, which was also valuable for designing new concrete bridges. However, it should be noted that the vibration energy of a bridge has a broadband frequency range since the speeds and weights of vehicles crossing the bridge change with time. Thus, the LPEH is inefficient in harvesting the broadband random vibration energy.

In recent years, some researchers tried utilizing nonlinearity to improve the operating bandwidth. The investigations on nonlinear energy harvesters with mono-stability [13], bi-stability [14], tri-stability [15] and quad-stability [16] have been carried out. For example, Stanton et al. [17] studied the hardening and softening response of the nonlinear mono-stable piezoelectric energy harvester. Sebald et al. [18] conducted a theoretical analysis of the nonlinear Duffing oscillator. The simulation results showed that the highest solution could obtain a huge gain in output power. Erturk and Inman [19] introduced a bi-stable piezoelectric energy harvester (BPEH) to improve the performance of power generation, which had a significant advantage over the LPEH. Arrieta et al. [20] introduced bi-stable characteristics in the cantilever composite to enhance the energy harvesting effectiveness under low-amplitude input levels. Masana et al. [21] designed an axially loaded BPEH, which could produce large output voltages. Lan et al. [22] added a small magnet in BPEH to improve the energy harvesting ability. Pellegrini [23] introduced a guideline to classify the BPEHs according to the feasibility of fabrication. Zhou et al. [24] proposed a broadband tri-stable piezoelectric energy harvester to further improve the energy harvesting ability. The results demonstrated that the tri-stable piezoelectric energy harvester could be superior to the bi-stable one, which was more suitable to the low input acceleration level. Litak et al. [25] investigated multiple solutions of a tri-stable energy harvester. It was found that the initial condition played an important role in the formation of the basins of attraction. Zhou et al. [26] designed a quad-stable energy harvester to improve the energy harvesting efficiency. The results proved the energy harvester designed in a quad-stable state made it much easier to attain snap-through motion, and its energy harvesting efficiency was significantly improved. Therefore, introducing nonlinear multi-stability is an effective way for efficient piezoelectric energy harvesting from the excitation in the shaking table test. Recently, Zhou et al. [27] exploited the mono-stable characteristics generated by the magnetic interaction to harvest bridge vibration energy, which brought about a noticeable improvement in the harvesting performance. However, to the authors’ best knowledge, the concept of harvesting energy from bridge vibrations by bi-stable characteristics has not been reported so far. In bridge vibration energy harvesting, the bi-stable characteristics may help improve the harvesting performance significantly.

In this paper, the bi-stable characteristics generated by the magnetic interaction were exploited to enhance the performance of energy harvesting from a bridge under the excitation of moving vehicles. When the excitation triggers the BPEH to execute snap-through motion and oscillates in a high-energy orbit, it can generate a large output. In order to study the dynamic characteristics of the BPEH, the potential energy, restoring force and stiffness are analyzed. Then, the output voltages and dynamic strains of LPEH and BPEH are measured in experiments and compared, as a vehicle model is designed to run on the bridge with different speeds. Finally, some important findings and conclusions are drawn.

## 2. Energy Harvester for Bridge Vibration Induced by a Moving Vehicle

The schematic sketch of the BPEH is shown in Figure 1. The BPEH is composed of a substrate partly covered by a piezoelectric patch or two, a tip magnet and two fixed magnets. A tip magnet is placed at the free end of the substrate, while two fixed magnets are mounted on the frame and positioned near the tip magnet. The tip magnet and fixed magnets have opposite polarities and face each other to produce a nonlinear magnetic attraction force. For clarity, we define two distances: one is the gap distance between two fixed magnets (*d_g_*); the other is the separation distance between the tip and the fixed magnets (*d*). Both of them can be adjusted to make the BPEH own two stable equilibrium positions. When the BPEH is excited by bridge vibrations, the piezoelectric patch will deflect and generate voltage. There is load resistance in the circuit. The BPEH can jump between the two stable equilibrium positions and execute snap-through motions, and then it will generate high voltage outputs. If the two fixed magnets are removed, the system degenerates to a classical LPEH. For the LPEH, as a linear system, it usually owns a narrow resonant bandwidth. Because the acceleration spectrum of bridge vibration usually exhibits a random characteristic and distributes over a wide range, the energy harvesting efficiency of LPEH under this excitation will be far from satisfactory.

Figure 2 shows the schematic of harvesting vibration energy from a bridge with a moving vehicle. As the vehicle runs on the bridge, the bridge is under the excitation of a moving load, so it will oscillate. In analysis, for simplicity, the vehicle can be modeled as a mass traveling on an elastic beam with length $L_{b}$ and thickness $T_{b}$ [9]. When the vehicle moves at a speed of *v* and passes the bridge, the bridge’s oscillation will make the BPEH vibrate and produce snap-through motions.

## 3. Modeling Potential Energy, Restoring Force and Stiffness

As we know, the LPEH has a linear restoring force and constant stiffness. It could exhibit a satisfactory work efficiency only when the frequency of the excitation source coincides with the resonance frequency. In fact, the vibration energy in the realistic environment usually distributes over a broadband frequency range, so the LPEH is inefficient in harvesting vibration energy. In contrast, the BPEH may overcome this defect.

To show the characteristics of the BPEH, its potential energy, restoring force and stiffness are presented. The total potential energy of the system includes three parts: the elastic potential energy of the substrate and piezoelectric patch, the gravity potential energy and the magnetic potential energy.

The substrate can be modeled as a Euler–Bernoulli beam, whose strain is proportional to the second spatial derivative of deflection. Thus, the elastic potential energy of the piezoelectric beam can be given by [28]
(1)$$U_{b}=\frac{1}{2}E_{s}I_{s}\int_{0}^{L_{s}}{\left(\frac{\partial^{2}y\left(x,t\right)}{\partial x^{2}}\right)}^{2}dx+E_{p}I_{p}\int_{0}^{L_{p}}{\left(\frac{\partial^{2}y\left(x,t\right)}{\partial x^{2}}\right)}^{2}dx$$
where $E_{s}$ and $E_{p}$ are the Young’s modulus of the substrate and that of the piezoelectric patch, respectively; $I_{s}$ and $I_{p}$ are the inertial moments of the substrate and the piezoelectric patch, respectively; $L_{s}$ and $L_{p}$ are the lengths of the substrate and the piezoelectric patch, respectively; y(x,t) is the displacement of the piezoelectric beam.

The magnetic potential energy of the system comes from the magnetic attraction between the tip and fixed magnets, and the point dipole approximation is employed to model the magnets. Then, the magnetic potential energy considered in the BPEH can be given by [29]:(2)$$U_{m}=-\frac{\mu_{0}a_{1}a_{2}}{2\pi }{\left\{{\left(y\left(L_{s},t\right)-\frac{d_{g}}{2}\right)}^{2}+d^{2}\right\}}^{-\frac{3}{2}}-\frac{\mu_{0}a_{1}a_{3}}{2\pi }{\left\{{\left(y\left(L_{s},t\right)+\frac{d_{g}}{2}\right)}^{2}+d^{2}\right\}}^{-\frac{3}{2}}$$
where $\mu_{0}$ is the magnetic permeability constant; $a_{1}$ and $a_{2}$ ($a_{3}$) are the effective magnetic moments of the tip and fixed magnets, respectively; d is the distance between the tip magnet and the fixed magnets in the horizontal direction (as shown in Figure 1); $d_{g}$ is the vertical distance between the two fixed magnets.

The gravitation potential energies of the substrate, piezoelectric patch and tip magnet can be given by the following equation:(3)$$U_{g}=-m_{t}gy\left(L_{s},t\right)-\rho_{s}h_{s}b_{s}g\int_{0}^{L_{s}}y\left(x,t\right)dx-2\rho_{p}h_{p}b_{p}g\int_{0}^{L_{p}}y\left(x,t\right)dx$$
where $m_{t}$ is the mass of the tip magnet; g is the gravitational acceleration; $\rho_{s}$ and $\rho_{p}$ are the densities of the substrate and piezoelectric patch, respectively;$\;h_{s}$ and $h_{p}$ are the thicknesses of the substrate and piezoelectric patch, respectively; $b_{s}$ and $b_{p}$ are the widths of the substrate and piezoelectric patch, respectively.

Thus, the total potential energy of BPEH can be given as follows
(4)$$U_{t}=U_{b}+U_{m}+U_{g}$$

The dynamic characteristics of the energy harvester can be inferred from its potential energy shape. Figure 3 shows the potential energy shapes of the harvester for different separation distances (*d*). The corresponding parameters are listed in Table 1. As is clear from Figure 3, when *d* is quite large (e.g., *d* = 90 mm), the harvester acts like a linear system since the magnetic force generated by the magnetic attractive effect is weak. Then, as *d* decreases (e.g., *d* = 17 mm), it turns into a mono-stable one due to the introduction of the magnetic force. It can be found that the curve of the potential energy function has a potential well and is asymmetric due to the gravitation effect. When *d* is less than a critical value, the attractive magnetic force will be greater than the resultant force of the elastic force and gravitational force. Thus, a potential barrier and two asymmetric potential energy wells are formed in the potential energy diagram. The asymmetry will become more obvious with the decrease in *d*. Subsequently, as *d* decreases further (e.g., *d* = 14 mm), the depths of two potential energy wells will increase, forming two deeper potential wells. For the bi-stable system, its snap-through motion may be excited to happen, which can make the piezoelectric beam have a large deflection and thus generate a relatively large voltage output.

To further study the characteristics of the BPEH, Figure 4 shows the restoring force and stiffness of the BPEH with respect to different separation distances (*d* = 14, 15, 16, 17, 18 or 90 mm). As *d* decreases, the equilibrium position corresponding to the zero value of the restoring force gradually shifts toward the negative direction due to the gravitation effect, as shown in Figure 4a. When the attractive magnetic and elastic forces get to the same order of magnitude, the coupling effect becomes stronger. Now, the gravity effect plays an increasingly important role. As the separation distance decreases, it will influence the equilibrium position of tip deflection, which corresponds to the point of zero restoring force. When there appear three zero points in the restoring force curve, the system exhibits bi-stable behavior. In particular, by adjusting *d*, the restoring force and stiffness shown in Figure 4 can be reduced to a very small value and even become negative. The restoring force in the bi-stable state is extremely low and exhibits a negative value near zero tip deflection, which is expected to give rise to a large deflection and generate a high output voltage under weak vibration of the bridge.

## 4. Experimental Setup

To validate the energy harvesting efficiency of the BPEH, corresponding experimental tests were carried out. The experimental setup is shown in Figure 5. The vehicle is represented by a steel ball (1003 kg). The model bridge used for experimental tests is made from an acrylic sheet (1300 mm × 80 mm × 8 mm), as shown in Figure 5a. Two guardrails are attached to the bridge to prevent the ball from falling off the bridge. Because the first mode plays an important role in bridge vibration, the first modal displacement is a determinant element for the installation of the harvester. For collecting high electric energy, the BPEH is installed under the midpoint of the bridge to harvest the vibration energy. As for the excitation, a steel ball is released from different heights and rolls down the acceleration section (inclined track). Then, it passes the bridge, as shown in Figure 5b. The prototype of the BPEH is fabricated, as shown in Figure 6. The two stable equilibrium positions of the BPEH are shown in Figure 7. The substrate is made of stainless steel and has dimensions of 190 mm × 10 mm × 1 mm. The piezoelectric patch has the dimensions of 5 mm × 5 mm × 0.25 mm, which is connected to a high-precision resistance box. A strain sensor (120–5 AA) is bonded on the substrate to measure the dynamic strain. As for the magnets, three permanent magnets have the same dimensions of 10 mm × 10 mm × 5 mm. If the two fixed magnets are removed, the BPEH will degenerate to a classical LPEH (as shown in Figure 8). In all experimental tests, the LPEH and BPEH are put in the same experimental conditions to make a direct comparison. A data acquisition device (DH5922N, Donghua, JingJiang, China) is adopted to measure the output voltage and dynamic strain. The schematic diagram of the experimental procedure is illustrated in Figure 9.

## 5. Results and Discussions

The dynamic behavior and electrical output are selected as the key indicators in evaluating the dynamic behavior and output performance. The steel ball is released from a titled track, which is named the acceleration section. In the acceleration section, for the ball, different positions could produce different initial speeds to pass the bridge. In the experiment, eleven positions are chosen to obtain different initial speeds. The running interval of the steel ball for measurement is set to be from 0.52 s to 1.18 s, which is consistent with the actual interval of a vehicle traveling through a bridge. Figure 10 shows the variance of strain and the power density (the load resistance is about 0.9 MΩ) of LPEH and BPEH for different moving speeds. It is clear from Figure 10 that, for BPEH, the variances of strain increase greatly at the speed of 1.68 m/s, resulting in a significant increase in the output power density. This phenomenon indicates that the BPEH is triggered to execute the snap-through motion and oscillates in a high-energy orbit. Thus, the BPEH can significantly improve the energy harvesting performance compared to the LPEH. For example, at the speed of 2.07 m/s, the BPEH’s maximum power density can reach 430 W/m^3^, whereas the LPEH’s maximum RMS voltage is only 261 W/m^3^.

To reveal the vibration response of the LPEH and BPEH, Figure 11 illustrates the time histories of strain for six moving speeds (*v* = 1.10, 1.52, 1.82, 2.07, 2.30 and 2.50 m/s). In the time response, the tag “On bridge” indicates the instant when the vehicle enters the bridge, while the tag “Leaving bridge” indicates the instant when the vehicle gets off the bridge. At a very low moving speed *v* = 1.10 m/s or 1.52 m/s, the BPEH’s oscillation cannot cross the potential barrier and thus is trapped in a potential well. Thus, both BPEH and LPEH have a small vibration amplitude now. When the moving speed increases to 1.82 m/s (Figure 11c), the BPEH produces a large-amplitude vibration, for it begins to realize jumps between two stable equilibrium positions. Now the maximum strain of the BPEH can reach 4.5 × 10^−4^, whereas the maximum strain of the LPEH is only 3.4 × 10^−4^. The amplitude of BPEH is larger than that of LPEH, and the vibration period of BPEH is longer as well. During the running period on the bridge, for the BPEH, the snap-through motion keeps happening. Finally, as the vehicle gets off the bridge, the response will fall into a potential well and oscillate in it. As the moving speed increases to 2.07 m/s, the vibration amplitude of BPEH will have a significant increase, whose maximum strain will increase up to 9.5 × 10^−4^ (Figure 11c). If the moving speed increases further to a high one (e.g., 2.30 m/s or 2.50 m/s), the frequent snap-through motion will happen in the BPEH, resulting in a much larger amplitude than the LPEH, as shown in Figure 11e,f. For example, at 2.30 m/s, the maximum strain of BPEH can reach 1.16 × 10^−3^, nearly two times that of the LPEH (6.8 × 10^−4^). Thus, through comparing the time histories of the strain of two systems, it can be concluded that the BPEH is preferable to the LPEH in producing large deformations and giving large outputs for the same running vehicle excitations.

Figure 12 shows the open-circuit output voltages of LPEH and BPEH for different moving speeds. It is well known that the PZT output is proportional to its deformation or strain. As is clear from Figure 12, the BPEH is superior to the LPEH in open-circuit voltage output at every speed. Specifically, at the low moving speed, e.g., *v* = 1.10 m/s, as Figure 12a shows, although the oscillation of BPEH is trapped in a potential well, the open-circuit voltage output of BPEH is still larger than that of LPEH due to the tailored potential energy shape. Then, when the moving speed increases to 1.82 m/s (Figure 12c), the BPEH’s maximum open-circuit output voltage has a large increase and reaches up to 4.09 V, for the snap-through motion happens now, whereas the LPEH’s maximum open-circuit voltage is only 2.44 V. Next, when the moving vehicle runs at a speed of 2.07 m/s, the open-circuit output voltage is further improved and is up to 6.72 V. However, if the moving speed increases continuously, the open-circuit output voltage of BPEH will not increase greatly, even will decrease. For example, as the moving speed reaches 2.30 m/s, the peak voltage of BPEH is 7.73 V. Then, as the moving speed increases a little to 2.50 m/s, the peak voltage is 7.56 V, a little smaller than that at 2.30 m/s. This is due to the protective effect of the fixed magnets. If the cantilever is excited to execute a very large vibration, the cantilever is likely to be damaged rapidly. Therefore, the attractive force between the tip and fixed magnets will take effect and constrain the excessively large vibration to protect the BPEH.

In order to show the characteristics of the open-circuit voltage response of LPEH and BPEH, corresponding frequency spectra for different moving speeds are shown in Figure 13. It can be found that the peak of the LPEH is located near 14.3 Hz for any speed of the moving vehicle, implying that it is the system’s resonance frequency. However, for the BPEH, the distribution of frequency spectra is different depending on the moving speed. At a relatively low moving speed (e.g., *v* = 1.10 m/s), the peak of the spectrum of BPEH output is located at 8.5 Hz, the BPEH’s amplitude is much larger than that of the LPEH. When the moving speed increases to 1.82 m/s, the main peak shifts to 5.2 Hz, and the frequency energy of BPEH distributes over a wide range of 0–11.4 Hz. This is because the BPEH executes snap-through motions and exhibits a strong nonlinear characteristic. As the moving speed continues to increase, the spectrum of BPEH distributes more widely in a frequency range of 0~17.2 Hz (as shown in Figure 13d,f). Therefore, it can be concluded that the BPEH has a wider bandwidth and exhibits strong nonlinearity, which could help harvest more energy from bridge vibration.

## 6. Conclusions

In this work, a bi-stable PEH was developed by introducing a tip and two fixed magnets. The bi-stable characteristic of this BPEH is generated by the magnetic attractive interaction, which helps enhance the energy harvesting from bridge vibrations. For the BPEH, corresponding theoretical analysis and simulations were carried out, and the validation experiment was conducted. The results show that the BPEH can significantly improve the energy harvesting performance by realizing snap-through motions. Compared to the LPEH, the BPEH has a wide frequency spectrum, which varies with the moving speed. The voltage response of the BPEH outperforms that of the LPEH over the whole range of moving speeds. The BPEH could significantly increase the deflection under the same excitation, which is beneficial for harvesting more energy from bridge vibration induced by traveling vehicles. Compared to the classical designs, the BPEH is simple in structure and can be easily adjusted in size to obtain a satisfactory harvesting performance. This design may lead to the realization of an efficient and reliable energy harvester. However, there remain some aspects to be studied in future work. For example, the separation and gap distances can be optimized so as to make the BPEH realize snap-through motions more easily. In practice, multiple vehicles will run on the bridge simultaneously, the BPEH’s response and output under this excitation should be studied.

## Figures and Tables

**Figure 1 materials-15-02237-f001:**
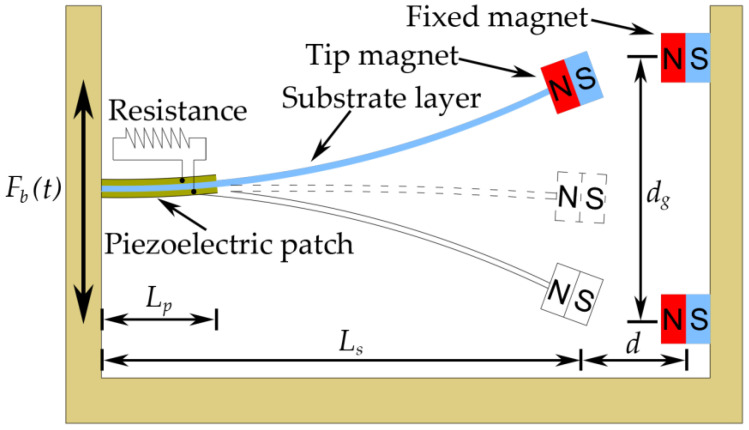
Schematic of the BPEH.

**Figure 2 materials-15-02237-f002:**
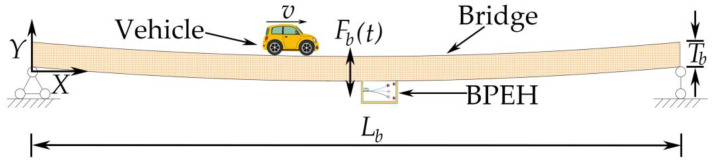
Energy harvesting from a bridge traversed by a vehicle.

**Figure 3 materials-15-02237-f003:**
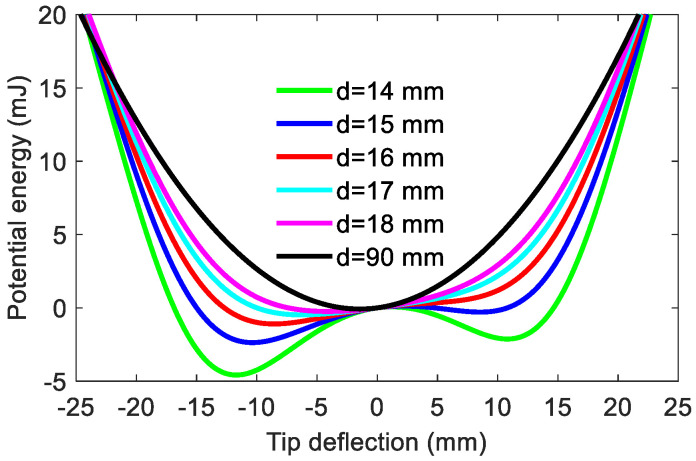
Potential energy shapes of the harvester for various separation distances.

**Figure 4 materials-15-02237-f004:**
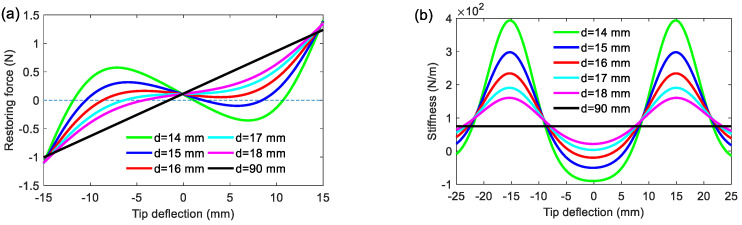
(**a**) Restoring force and (**b**) stiffness of the BPEH for various separation distances.

**Figure 5 materials-15-02237-f005:**
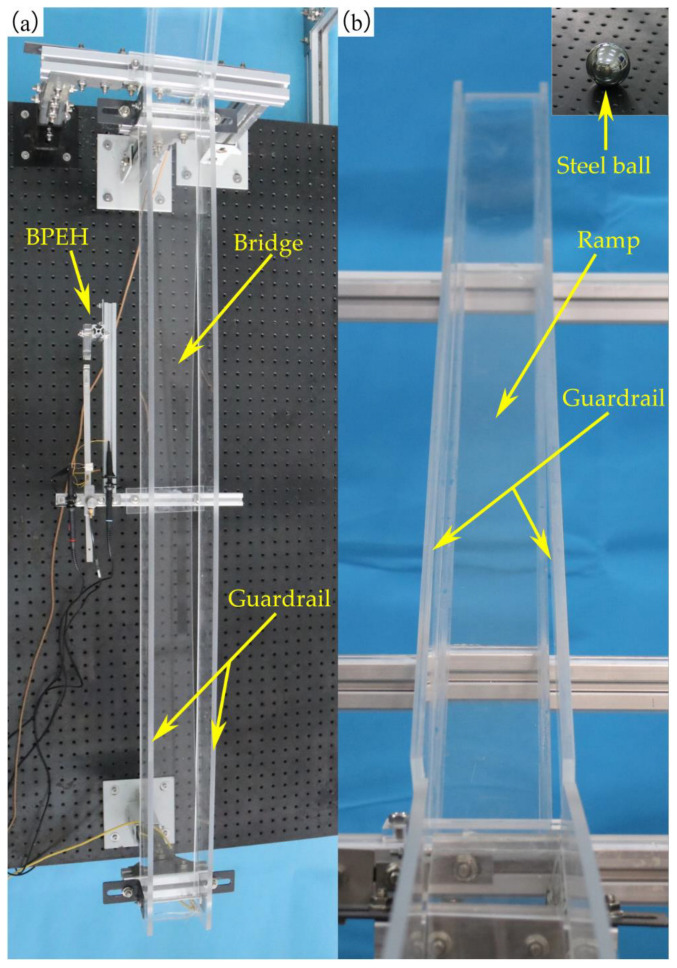
The experiment setup: (**a**) test section and (**b**) acceleration section.

**Figure 6 materials-15-02237-f006:**
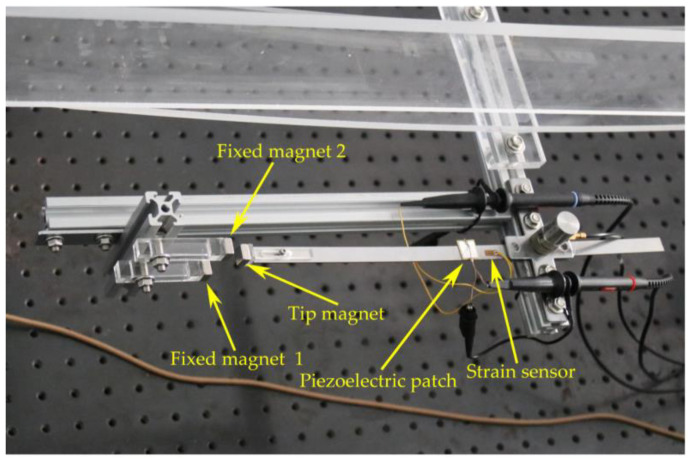
The prototype of the BPEH.

**Figure 7 materials-15-02237-f007:**
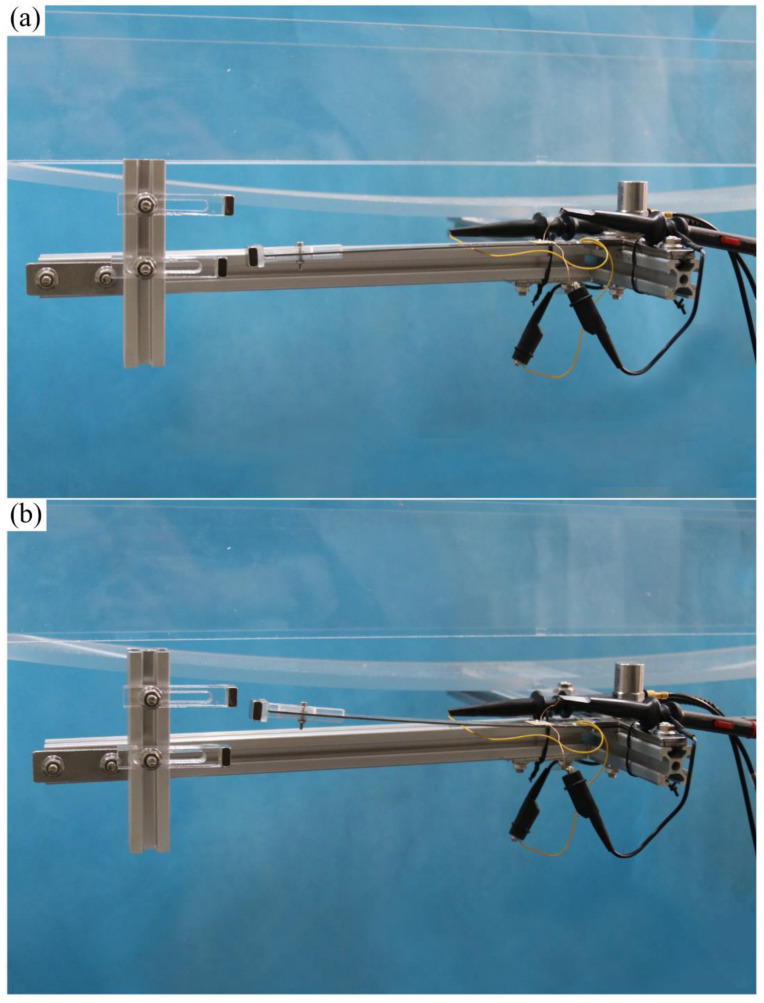
The two stable equilibrium positions of the BPEH: (**a**) stable equilibrium position 1 and (**b**) stable equilibrium position 2.

**Figure 8 materials-15-02237-f008:**
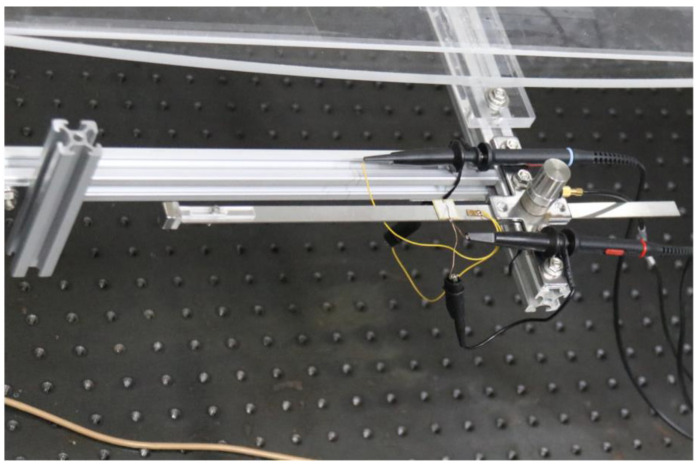
The prototype of the LPEH.

**Figure 9 materials-15-02237-f009:**
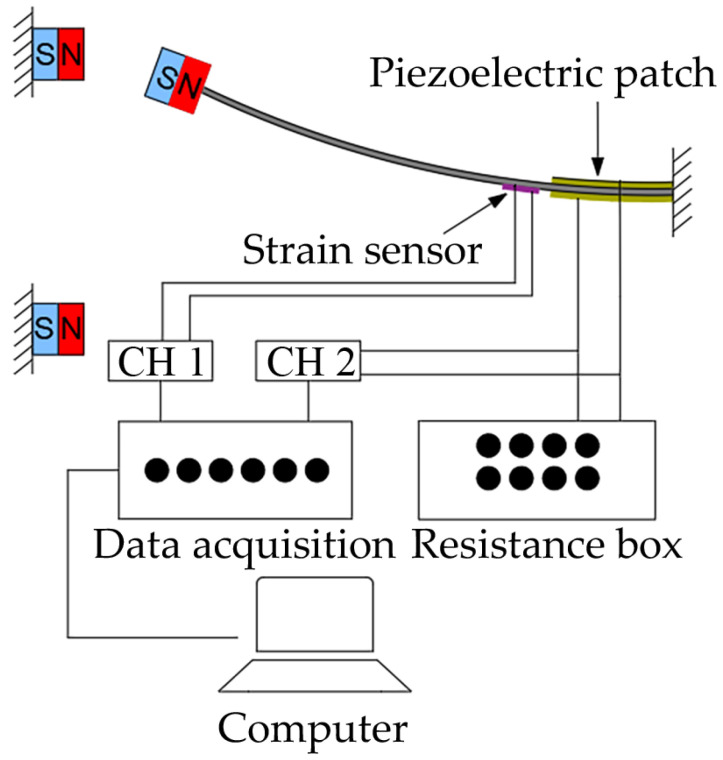
Schematic diagram of experimental procedure.

**Figure 10 materials-15-02237-f010:**
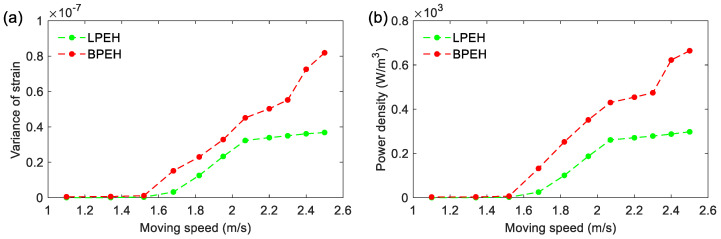
(**a**) Variance of strain and (**b**) power density versus moving speed for LPEH and BPEH.

**Figure 11 materials-15-02237-f011:**
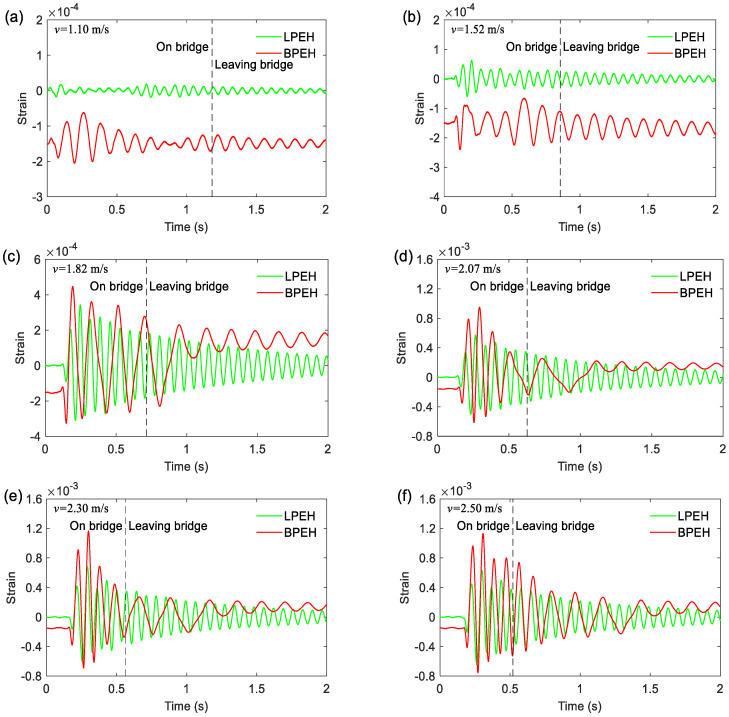
Strain responses of the LPEH and BPEH for the vehicle traveling on and leaving the bridge: (**a**) *v* = 1.10 m/s; (**b**) *v* = 1.52 m/s; (**c**) *v* = 1.82 m/s; (**d**) *v* = 2.07 m/s; (**e**) *v* = 2.30 m/s; and (**f**) *v* = 2.50 m/s.

**Figure 12 materials-15-02237-f012:**
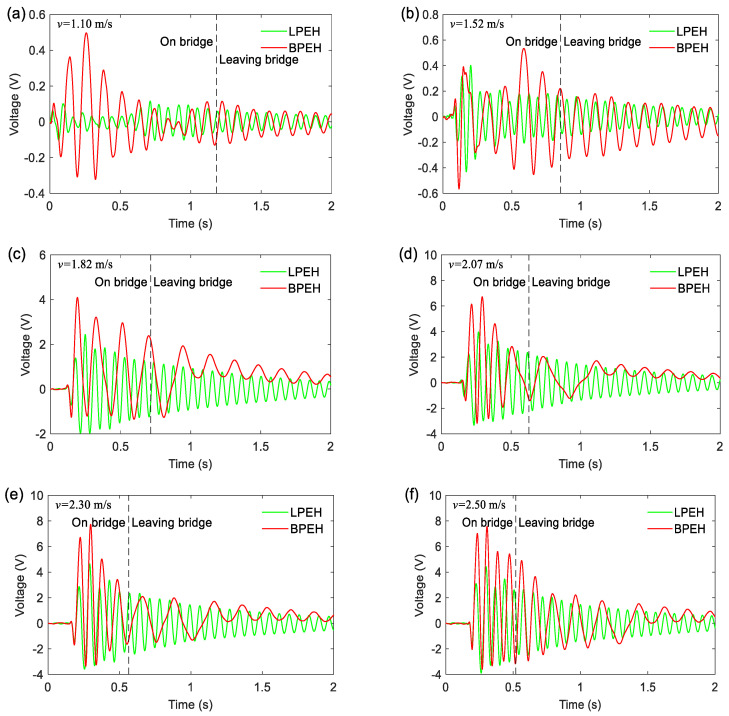
Open-circuit output voltages of the LPEH and BPEH for the vehicle traveling on and leaving bridge: (**a**) *v* = 1.10 m/s; (**b**) *v* = 1.52 m/s; (**c**) *v* = 1.82 m/s; (**d**) *v* = 2.07 m/s; (**e**) *v* = 2.30 m/s; and (**f**) *v* = 2.50 m/s.

**Figure 13 materials-15-02237-f013:**
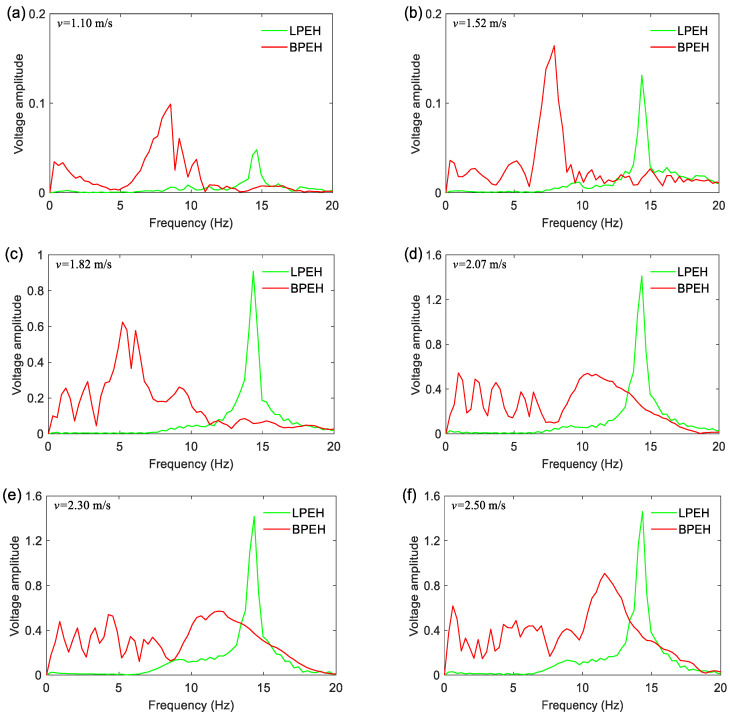
Frequency spectra of the open-circuit output voltages for the vehicle traveling on and leaving bridge: (**a**) *v* = 1.10 m/s; (**b**) *v* = 1.52 m/s; (**c**) *v* = 1.82 m/s; (**d**) *v* = 2.07 m/s; (**e**) *v* = 2.30 m/s; and (**f**) *v* = 2.50 m/s.

**Table 1 materials-15-02237-t001:** Parameters used for the analysis of potential energy function.

Parameters	Value
Substrate properties	
$\mathrm{Length}\;L_{s}$	190 mm
$\mathrm{Thickness}\;h_{s}$	1 mm
$\mathrm{Width}\;b_{s}$	10 mm
$\mathrm{Inertial}\;\mathrm{moment}\;I_{s}$	0.833 mm^4^
$\mathrm{Young}\;\mathrm{modulus}\;\;E_{s}$	205 GPa
$\mathrm{density}\;\rho_{s}$	7.8 g/cm^3^
Piezoelectric patch properties	
$\mathrm{Length}\;L_{p}$	5 mm
$\mathrm{Thickness}\;h_{p}$	0.25 mm
$\mathrm{Width}\;b_{p}$	5 mm
$\mathrm{Inertial}\;\mathrm{moment}\;I_{p}$	0.49 mm^4^
$\mathrm{Young}\;\mathrm{modulus}\;E_{p}$	56 GPa
$\mathrm{density}\;\rho_{s}$	2.7 g/cm^3^
Magnet properties	
$\mathrm{Effective}\;\mathrm{magnetic}\;\mathrm{moment}\;a_{1}\left(a_{2},a_{3}\right)$	$0.218\;Am^{2}$
$\mathrm{Mass}\;m_{t}$	3.7 g
$\mathrm{Gap}\;\mathrm{distance}\;\mathrm{between}\;\mathrm{two}\;\mathrm{fixed}\;\mathrm{magnets}\;d_{g}$	45 mm
$\mathrm{Permeability}\;\mathrm{constant}\;\mathrm{of}\;\mathrm{magnet}\;\mu_{0}$	$4\times 10^{-7}$ $\;NA^{-2}$

## Data Availability

The data presented in this study are available on request from the corresponding author.

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
