# Peer review of "Improving Energy Harvesting from Bridge Vibration Excited by Moving Vehicles with a Bi-Stable Harvester"

_materials, 2022, doi:10.3390/ma15062237_

Round 1

Reviewer 1 Report

The reviewer appreciates the theoretical and experimental work done by the author. The technical contents of the paper are in general interesting, and promising for bridge applications. Further investigations should focus on laboratory experiments in which the bridge prototype has larger geometric dimensions. Moreover, in the reviewer’s opinion, the goal of the work must be better explained within Abstract, Introduction and Conclusions. So, the publication in the “Materials, MDPI” is recommended if the following suggestions are taken into account within the article:

1) The proposed device can have a great effectiveness for long term monitoring of bridges. Such test monitoring usually requires the measurement of fundamental frequency of the bridge (or parts of it) using ambient and/or free vibrations. Conventional sensors are often utilized. The stiffness of the bridge is then estimated. Please, mention about this issue within the Introduction, and cite the following corresponding references:
-  Experimental–theoretical investigation of the short-term vibration response of uncracked prestressed concrete members under long-age conditions. Structures 2022 35 pp. 260–273.
-  Indirect assessment of concrete resistance from FE model updating and Young's modulus estimation of a multi-span PSC viaduct: Experimental tests and validation. Structures 2022 37 pp. 686–697.

2) Additional comments should emphasize the practical value of this work, and how the industry can profit from this article.

3) Fig. 1: “Schematic of the BPEH”. Please, describe more clearly the single parts of the device proposed within the text.

4) I suggest to the authors to edit all the text of the paper with the help of a native English speaker. Grammar, punctuation, spelling, verb usage, sentence structure, conciseness, readability and writing style could be improved.

Reviewer 2 Report

The article concerns the interesting topic of autonomously powered bridge vibration sensors. The issue is well described, clearly explained and illustrated. Good interpretation of the results. Benefits are indicated.

However, I have a few comments:

The previous article by the authors has already been published: doi.org/10.3390/ma15010033. It can be located in the state of knowledge.
It is also advisable to supplement in line 45 doi.org/10.3390/en14051284.
The analysis of the measurement results is appropriate. It is advisable to supplement or present phase portraits or stroboscopic maps in the next article. Therefore, in the state of knowledge, it is worth mentioning doi.org/10.1177/1045389X12444940

I propose to give an explanation: The bridge model bends in the first mode. How does this relate to the real bridge?

Round 2

Reviewer 1 Report

The authors have adequately addressed my comments.